

# Coherent diffraction imaging for enhanced fault and fracture network characterization

Benjamin Schwarz[1] and Charlotte M. Krawczyk[1,2]

[1]GFZ German Research Centre for Geosciences, Albert-Einstein-Str. 42-46, 14473 Potsdam, Germany
[2]Technical University Berlin, Ernst-Reuter-Platz 1, 10589 Berlin, Germany

**Correspondence:** Benjamin Schwarz (bschwarz@gfz-potsdam.de)

**Abstract.** Faults and fractures represent unique features of the solid Earth and are especially pervasive in the shallow crust. Aside from directly relating to crustal dynamics and the systematic assessment of associated risk, fault and fracture networks enable the efficient migration of fluids and, therefore, have a direct impact on concrete topics relevant to society, including climate-change mitigating measures like $CO_2$ sequestration or geothermal exploration and production. Due to their small-scale
complexity, fault zones and fracture networks are typically poorly resolved and their presence can often only be inferred indirectly in seismic and ground-penetrating radar (GPR) subsurface reconstructions. We suggest a largely data-driven framework for the direct imaging of these features by making use of the faint and still often under-explored diffracted portion of the wavefield. Finding inspiration in the fields of optics and visual perception, we introduce two different conceptual pathways for coherent diffraction imaging and discuss respective advantages and disadvantages in different contexts of application. At the
heart of both of these strategies lies the assessment of data coherence, for which a range of quantitative measures is introduced. To illustrate the approaches versatility and effectiveness for high-resolution geophysical imaging, several seismic and GPR field data examples are presented, in which the diffracted wavefield sheds new light on crustal features like fluvial channels, erosional surfaces, and intricate fault and fracture networks on land and in the marine environment.

## 1   Introduction

Crustal faults and fracture systems are of significant importance for the structural interpretation of geophysical images. Resulting from acting forces they not only encode past configurations of local stress fields, but also represent primary indicators of man-made or natural hazards or fluid flow in the subsurface (Sibson, 1994). In addition, the delineation of faults also helps to shed light on the mechanical properties of the host material and provides valuable assistance in tracking horizons and spatially linking stratigraphic units in sedimentary regimes. Crystalline-rock environments, which are of special interest for geothermal
exploration and production, are known to be brittle and scarred by intricate fracture networks, whose successful identification and characterization has an immediate impact on the desired transition to sustainable energies. Despite their importance, pronounced direct geophysical images of crustal faults, in particular when temporarily inactive, remain largely elusive, owing in large parts to their sub-wavelength structural complexity and the seemingly diffuse and complex wavefields that are typically associated with them.



With a long history in optical imaging, the wave process of diffraction is synonymous with the highest possible resolution achievable in a reconstruction (Born and Wolf, 2013). Large parts of the Earth's crust are known to heavily diffract incoming seismic or electromagnetic radiation. However, exploration and earthquake seismology either rely on transmitted, reflected and converted arrivals or surface waves and often implicitly ignore weaker, seemingly uncorrelated contributions for the direct imaging of the subsurface. Constrained by the interference with other typically stronger reflected or transmitted phases, individual

diffractions are often hard to identify on isolated records, despite the fact that they represent coherent signal. The suitability of seismic diffractions as a direct fault indicator was already explored in the 1950s and was further investigated in the following two decades (Krey, 1952; Kunz, 1960; Trorey, 1970; Berryhill, 1977). While these studies were mostly concerned with the accurate numerical modelling of the diffraction response, the first imaging attempts, despite their novelty, largely suffered from inadequate data quality (Landa et al., 1987; Kanasevich and Phadke, 1988). After an extended period in which seismic

migration and waveform inversion techniques evolved to their current sophistication (Etgen et al., 2009; Virieux and Operto, 2009), advancements in data acquisition led to a recent re-discovery of diffraction imaging for geophysical applications.

Coherence is a collective property of a wavefield and can be viewed as a pre-requisite for migration-type imaging. Recent decades have proven the usefulness of systematically assessing this property for applications like noise suppression (Mayne, 1962), wavefront attribute extraction (Gelchinsky et al., 1999a, b; Jäger et al., 2001), data interpolation and regularization

(Baykulov and Gajewski, 2009; Hoecht et al., 2009), wavefield separation (Bergler et al., 2002), velocity inversion (Symes and Carazzone, 1991; Billette and Lambaré, 1998; Duveneck, 2004), or passive-source localization (Schwarz et al., 2016; Diekmann et al., 2019). With the increasing availability of dense acquisition systems, different forms of coherence arguments have been invoked in seismic and ground-penetrating radar (GPR) diffraction imaging. Arguably one of the most important applications and stumbling blocks for successful high-resolution imaging is the separation of the faint diffracted wavefield

from stronger, often heavily interfering contributions. While some approaches introduced a diffraction bias in the migration scheme (Khaidukov et al., 2004; Moser and Howard, 2008; Klokov and Fomel, 2012), other strategies aim at extracting the weak diffraction response in a separate step before imaging (e.g. Bansal and Imhof, 2005; Fomel et al., 2007).

Likewise applied before migration, there exist techniques that make direct use of wavefield coherence for diffraction separation (Berkovitch et al., 2009; Dell and Gajewski, 2011; Bauer et al., 2016; Bakhtiari Rad et al., 2018). While these methods

specifically target the diffracted wavefield for extraction, recent developments have shown that a more surgical, amplitude-preserving separation can be achieved by assessing the coherence of reflections instead (Schwarz and Gajewski, 2017a; Schwarz, 2019b). Although other methods like e.g. plane-wave destruction can achieve a similar quality of extraction in many applications, the systematic and physically intuitive assessment of coherence can be carried out in any imaginable data configuration and allows for a seamless integration of data enhancement, wavefield separation, and imaging into a single framework.

Recent studies suggest that, owing to their unique properties, diffractions also lend themselves well for velocity inversion (Sava et al., 2005; Fomel et al., 2007; Decker et al., 2017; Bauer et al., 2017). These approaches bear the potential for a self-contained imaging and inversion cycle that is also applicable in the case of offset-limited acquisitions as they can often be found in academia (Preine et al., 2020).





With only few exceptions (e.g. Landa et al., 1987; Heincke et al., 2006; Dell et al., 2019), the potential of quantitative
coherence measures for directly forming noise-robust, contrast-rich images remains largely unexplored. Building on recent
advances in adaptive processing and weak-wavefield enhancement, we present a strategy for seismic and GPR diffraction
imaging that makes direct use of wavefield coherence for scatterer detection. After a brief elaboration on typical characteristics
and unique properties of diffraction phenomena, we introduce two different means of reconstructing a scatterer with the help of
coherence measurements. Underpinning both these pathways we introduce generalized coherence measures and systematically
investigate their tolerance with respect to imperfect, i.e. noisy, sparse, or incomplete data and make suggestions with respect to
their applicability. Concluding community-spanning seismic and electromagnetic examples suggest that coherent diffraction
imaging not only leads to overall highly-resolved subsurface reconstructions, but also directly and reliably highlights small-
scale erosional features, faults and fractures.

## 2 Wave diffraction

Diffraction can loosely be defined as a waves ability to enter *shadow zones*, which are forbidden regions in geometrical optics.
More precisely, diffraction occurs when a wavefield encounters a relevant property change that has a local curvature of or
below the wave length (Born and Wolf, 2013). Thus, diffraction is a scale-spanning phenomenon that is only predicted and
fully captured in a wave theoretical framework. To provide some intuition, Figure 1 illustrates some of the key properties of
diffractions and how they can be of use for geophysical subsurface imaging (for more details, see Schwarz, 2019a). As arguably
the first rigorous experimental evidence, Young's slit experiments concluded that when light hits a small enough opening in a
screen, an intricate interference pattern appears on a second screen. The geophysical analogue of such an experiment is shown
in Figure 1(a), where a small-scale heterogeneous object or a gap at an interface acts as a *secondary source*. The acquisition
surface at zero depth can be viewed as a screen, where the wavefields are captured by seismometers or electromagnetic anten-
nas. The mere occurrence of such a secondary wavefield (for clarity the primary field is not displayed) is already indicative of
the presence of a small-scale structural change underneath our acquisition surface, which is why it is frequently suggested, that
the detection and localization of such a structure potentially implies super-resolution imaging, i.e. the inference of structural
features of spatial extent beyond the Rayleigh limit (Khaidukov et al., 2004).

Because their wavefronts are principally indistinguishable from an actual source located at the scatterer location, seismic
diffractions share striking similarities with passive sources (Li et al., 2020). In Figure 1(b) the surface projections of the
diffracted wavefield shown in (a) are displayed as a function of time. Generally, although this is not precisely true in realistic
media, diffractions have approximately hyperbolic shape. It is interesting to note that, in contrast to reflections and other
wavefield components obeying Snell's law, diffractions always have a similar shape, no matter which data configuration is
considered. This implies that diffractions not only provide improved illumination and encode highly resolved information on
the structures that caused them, but it also explains why diffracted signals are often an order of magnitude weaker than their
reflected counterparts. Closely related to diffraction is the concept of interference, which likewise is a pure wave phenomenon.
Interference is mentioned here for two reasons. First, it explains the transitional regime and provides a notion of resolvability




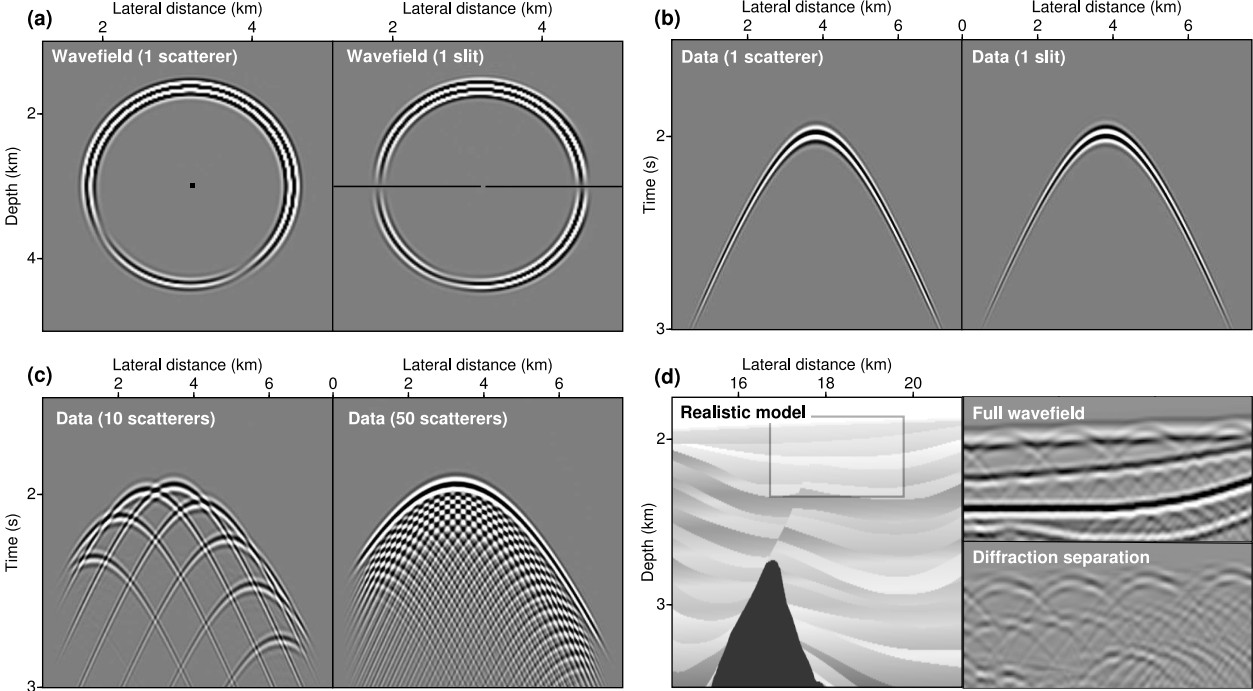

**Figure 1.** Illustration of the most important properties of diffractions. (a) Wave diffraction occurs if medium properties change with a local curvature comparable to the dominant wavelength (the primary wavefield causing the diffraction is not shown). (b) Recorded are surface projections as a function of time, which in mildly heterogeneous media are close to hyperbolic, independent of whether zero-offset, common-source, or common-receiver configurations are considered. (c-d) As manifestation of the Huygens-Fresnel principle the interference of infinitely many diffractions describe transmitted or reflected arrivals which honour Snell's law.

(Figure 1(c)), and second, it helps to appreciate the need and possibility for wavefield separation, in particular, when highly reflective media like sedimentary basins are considered (Figure 1(d)). When a sufficiently large amount of scatterers is present, individual diffractions become hardly distinguishable. Essentially all diffraction separation strategies rely on dense spatial

sampling at the surface. The high-resolution and high-illumination component of diffractions, which bear unique imaging potential, can only be unlocked, if spatial aliasing can be prevented (Schwarz, 2019a).

Aside from illustrating lateral resolvability, Figure 1(c) also visualizes the underlying principle of Kirchhoff migration (Schneider, 1978). As will be more thoroughly explained in the following section, Kirchhoff migration is a manifestation of the Huygens-Fresnel principle, which states that any arbitrary wavefield can be thought of as being composed of infinitely many

elementary wavefields. The envelope of these elementary, locally excited waves then forms the transmitted or reflected arrival. Diffractions can be viewed as physically resolved manifestations that are picked out by small-scale disturbances, as e.g. caused by faults. Figure 1(d) shows a small excerpt of a subsurface model that mimics the Sigsbee escarpment in the Gulf of Mexico. Aside from structural features related to the top of salt or a fault cutting through the sedimentary strata, wave diffraction is





also caused by stair stepping in the discretized model used for the finite-difference simulations. Despite their unintentional
introduction, the resulting pervasive diffracted wavefields nicely illustrate the transition from diffraction to reflection and why
a successful separation of these contributions remains a challenge to confront. High-resolution imaging aims at back-tracing
these weak diffracted contributions to their origin.

## 3   Coherent diffraction imaging

In conventional diffraction imaging, wavefield separation is either performed before or during migration (see, e.g., Fomel et al.,
2007; Moser and Howard, 2008). Independent of what type of migration algorithm is used, the result commonly comprises a
wavefield image that contains amplitude and phase information. While the preservation of phase information in the reconstruc-
tions is principally desirable, there exist several shortcomings of conventional migration schemes, in particular when imperfect
data and weak signals such as diffractions are concerned. In coherent diffraction imaging we seek to directly incorporate wave-
field coherence in the imaging workflow to help overcome these limitations. It is generally interesting to note that when optical
images are concerned, we only perceive intensities, and wavelength information is encoded in the colours of the visible light, to
which the eye is sensitive. Following this intuition, we argue that coherence measures, to some degree, mimic intensities and,
therefore, seem principally suited for the construction of structural images. As was briefly illustrated in the previous section,
diffractions are coherent and, just like reflections, can benefit from coherence arguments.

The imaging problem can generally be divided into two domains – the data and the image. Migration-type imaging, just like
an optical lens, seeks to directly utilize the former to arrive at the latter. In both, data and image space, coherence arguments
can be invoked (Figure 2). While the systematic assessment of coherence in data space has a long and successful history, using
coherence measures, with some few exceptions, have not been utilized to their full potential, when the image space is concerned.
With intuition from the field of optics, in order to properly differentiate between these two philosophies, we refer to starting
in data space, i.e. with the observations, as *projection*, whereas the image-centric approach will be denoted by *focusing*. Both
mindsets have in common that we use the data to construct an image and both are amenable to improvements when some form
of coherence measure is introduced. Consequently, *coherent focusing* evaluates wavefield coherence during the gathering stage
in image space, whereby *coherent projection* first evaluates data coherence and then back-projects with help of the extracted
information. Wave equation migration (focusing) and time-reversal imaging (projection) can be viewed as the most capable end
members of these two branches which find themselves powerfully combined in reverse-time migration for reflection imaging
(Baysal et al., 1983). While fully honoured wave propagation physics becomes important in sufficiently complex scenarios,
the unique flexibility of Kirchhoff migration and its intimate relation to wave diffraction provides unique opportunities for the
imaging of scatterers (Moser and Howard, 2008). Although wave propagation is abstracted by high-frequency approximations
of limited validity, the use of only kinematic information lets the developed framework be readily applicable for both, seismic
and GPR measurement campaigns.





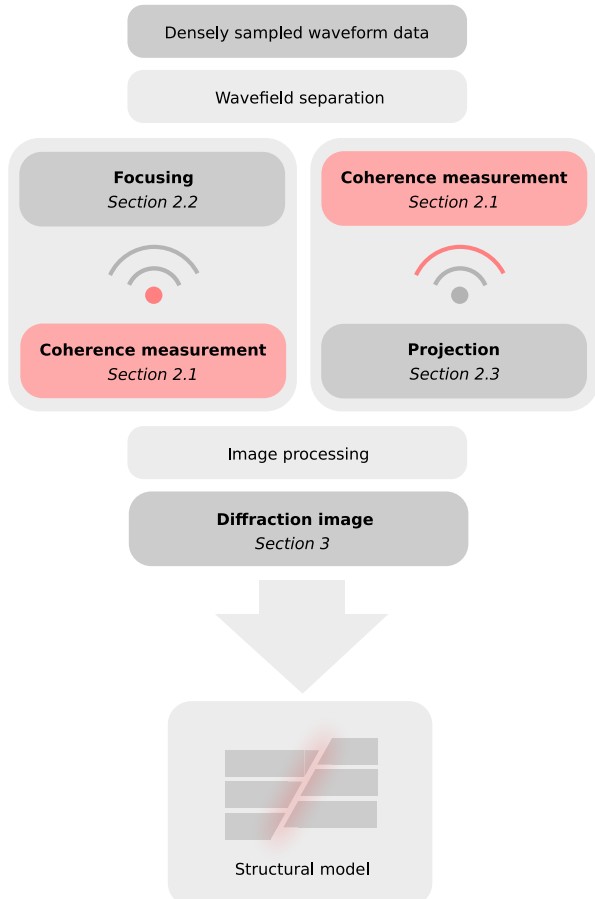

**Figure 2.** Generalized workflow for diffraction imaging. Indicated in red are instances, where coherence measurements enter the reconstruction. While projection-type imaging schemes start directly in data space, focusing techniques typically are image-centred. Although diffraction separation prior to imaging forms a central ingredient, emphasis is given to the reconstruction steps.

### 3.1 Measuring coherence

As illustrated in Figure 1, coherence is an observable contained in densely acquired waveform data. It can be assessed in depth (Figure 1(a)) and time (Figure 1(b)), i.e. in image and in data space. Coherence can be viewed as a set of correlations that are connected by the temporal or spatial delays arising from the shape of the propagating wavefront. If data are not acquired densely at the surface, these group correlations cannot be reliably tracked and connected any more. A well-appreciated way of assessing wavefield coherence is to perform directional data summations within a predefined time window. If we denote the spatio-temporal waveform data recorded at the surface (at position $\mathbf{x}$ and time $t$) by $\mathcal{D}(\mathbf{x}, t)$, we can write the summed




amplitude at point $\xi$, estimated within a confined aperture spanned by all $\mathbf{x}$, as

$$B(\xi) = \sum_{\mathbf{x}} \mathcal{D}[\mathbf{x}, t = t(\xi, \mathbf{x})] \tag{1}$$

where $t(\xi, \mathbf{x})$ is the traveltime surface corresponding to the wavefront. If this traveltime surface describes an actual event

(compare Figure 1(b)), the summation result $B(\xi)$ shows increased amplitudes, while for uncorrelated noise or a wrong or inaccurate choice of $t(\xi, \mathbf{x})$, the amplitudes are smaller. Equation (1) can be used to systematically suppress uncorrelated noise or undesired interfering coherent energy, but does not lend itself well for an automated analysis of wavefield coherence. In addition, the summed wavefield, like the data itself, encodes phase information resulting in positive and negative values, which complicates interpretation. In analogy to optics, we will refer to expression (1) as the *beam amplitude* or *beam*, which follows

the physically intuitive convention in earthquake seismology (Rost and Thomas, 2002). To arrive at a more robust quantity that can act as a cost function in an optimization scheme, the *beam energy* can be approximated as follows

$$E(\xi) \simeq \sum_{\tau} B^2(\xi), \tag{2}$$

where $\tau$ is a small time window in which vertical summation is performed. As a rule of thumb, it should have approximately the size of the considered signal's wavelength. The beam energy $E(\xi)$ takes only positive values, but does not precisely correspond

to the beam's energy but rather is proportional to it. In earthquake seismology, equation (2) is investigated routinely in slowness and back-azimuth analysis (Rost and Thomas, 2002). If we consider the total energy contained in the investigated portion of the wavefield, a similar proportionality holds

$$E_{\text{total}}(\xi) \simeq \sum_{\tau} \sum_{\mathbf{x}} \mathcal{D}^2[\mathbf{x}, t = t(\xi, \mathbf{x})]. \tag{3}$$

Expression (3) lets us arrive at an upper bound, as coherent summation has to honour energy conservation. The wavefields

*semblance*

$$S(\xi) = \frac{1}{N_{\mathbf{x}}} \frac{E(\xi)}{E_{\text{total}}(\xi)} \tag{4}$$

thus is a normalized quantity and was demonstrated to be an ideal candidate for the automation of coherence analysis (Taner and Koehler, 1969; Neidell and Taner, 1971). The quantity $N_{\mathbf{x}}$ indicates the total number of traces at all recording locations $\mathbf{x}$ that fall within the considered aperture. For perfectly coherent signal $S$ approaches 1, whereas for fully uncorrelated noise it

takes values close to 0. If data are very noise contaminated or other contributions interfere strongly with the event investigated, it can be useful to abstract the waveform data before processing (Li et al., 2020). One such abstraction constitutes the polarity-honouring $n$-th root of the signal

$$\mathcal{D}_n(\mathbf{x}, t) = \text{sgn}[\mathcal{D}(\mathbf{x}, t)] \sqrt[n]{|\mathcal{D}(\mathbf{x}, t)|}. \tag{5}$$

One main advantage of the $n$-th root abstraction is that in contrast to other means like kurtosis, the transformed data still retain

their polarity, which allows for destructive interference to occur. Insertion of expression (5) for $\mathcal{D}(\mathbf{x}, t)$ in equations (1)-(4) then



leads to $n$-th root versions of the beam amplitude $B_n(\xi)$, the beam energy $E_n(\xi)$, and semblance $S_n(\xi)$ (compare Schwarz, 2019a). For $n = 1$ all of these expressions reduce to their conventional analogues, which is why in the following, we will refer to different versions of each quantity by their order $n$. A systematic investigation of the above coherence measures will be carried out in the following two subsections. Nevertheless, it can already be stated that the beam energy (2) bears a strong

resemblance to a wavefields intensity, which likewise is sensitive to the absolute amplitude of a signal. As a consequence, it ascribes higher values to stronger, more energetic contributions (stronger scatterers appear *brighter*). Conversely, semblance represents an energy ratio and, owing to its normalization, coherence is detected independent of signal strength.

## 3.2 Imaging by focusing

As indicated earlier, imaging by focusing can conveniently be based on Kirchhoff's diffraction integral, which in practice, like

equation (1) reduces to a discrete sum. In a more physical sense focusing can be viewed as a special form of constructive interference, in which different measurements of the same coherent wavefield are superposed at the image location $\xi$. This image point represents either a sample in the spatio-temporal focussed image $\xi = (\mathbf{x}_0, t_0)$ corresponding to time migration, or fully spatial reconstruction with a depth axis $\xi = (\mathbf{x}_0, z_0)$ (Schneider, 1978; Etgen et al., 2009). As we are primarily interested in robust structural images of small-scale crustal features, here, in line with other authors (Khaidukov et al., 2004; Moser

and Howard, 2008), we neglect individual amplitude weighting which is normally applied in true-amplitude migration. As a consequence, the discrete version of the Kirchhoff integral is equivalent to the beam amplitude (1). Although all the presented findings likewise translate into depth, for the sake of simplicity and in order to stay consistent with the field data examples for which detailed velocity information was scarce, in the following, we will only consider spatio-temporal focusing.

In order to arrive at a full equivalence of the un-weighted Kirchhoff integral with the beam amplitude, for every image

point $\xi = (\mathbf{x}_0, t_0)$, the traveltime response of a diffraction needs to be inserted into equation (1). While for depth migration the accurate traveltimes of a diffracted wavefront are computed by means of ray tracing or eikonal solvers, time imaging typically relies on analytical, closed-form approximations whose validity is restricted to mild levels of lateral heterogeneity. A popular assumption is that traveltime moveout is approximately hyperbolic, which corresponds to circular wavefronts in an effective replacement medium (e.g. Schwarz and Gajewski, 2017b). This effective medium is described by the root-mean-square velocity

$v_{\text{rms}}$, which generally is a function of the image point. In compact notation, the multi-channel diffraction traveltime $t_{\text{diff}}$ can be written as a sum of a source and a receiver leg connected by the shared image point

$$t_{\text{diff}}(\mathbf{x}_0, t_0) = \sum_{i=s,g} \sqrt{\frac{t_0^2}{4} + \frac{\Delta\mathbf{x}_i^2}{v_{\text{rms}}^2(\mathbf{x}_0, t_0)}}, \tag{6}$$

where $\Delta\mathbf{x}_i = \mathbf{x}_i - \mathbf{x}_0$. Figure 3 illustrates with a synthetic example, how the previously discussed coherence measures compare for high-quality, i.e. densely and regularly sampled data with a low noise level, severe noise contamination, data sparsity

resulting in spatial aliasing, and incomplete (single-sided) observations. Without loss of generality, alongside the un-weighted Kirchhoff reconstruction, i.e. the beam amplitude (1) with $t = t_{\text{diff}}$, only the first and the 10-th order of the beam energy (2) and the semblance coefficient (4) is displayed. For the high-quality data (top row in Figure 3) all considered coherence measures





**Figure 3.** Diffraction imaging results gained through focusing with the conventional Kirchhoff-type beam amplitude and four quantitative coherence measures. Displayed are the results for ideal, noisy (signal-to-noise ratio 1), sparse (every fifth trace) and incomplete (left third of the line) data in which a strong residual reflection event is present. Desirable is an image, in which the positions of the two diffractors are well-determined and the reflected energy is maximally suppressed.

arrive at an accurate reconstruction of the two scatterers, while in the case of data insufficiencies especially the conventional Kirchhoff-type reconstruction with the beam amplitude suffers from strong imaging artefacts. In Kirchhoff migration, diffrac-

tions are naturally favoured in that in contrast to reflections, the summation trajectory is not only tangential, but fully coincides with the event. Because the two diffractors are located in different depths and at different lateral positions, they are not equally well imaged in all scenarios. Especially when only the left-sided incomplete observation is concerned, the right diffractor remains poorly resolved with the beam amplitude, the conventional beam energy, and the conventional semblance norm.

    As a typical challenge in diffraction imaging constitutes the presence of residual reflected energy after separation, we in-

cluded the response of a planar reflector in the deeper part of the model. While this strong but undesired contribution is only mildly suppressed in the conventional Kirchhoff reconstruction, the coherence images are mostly reflection-free with only



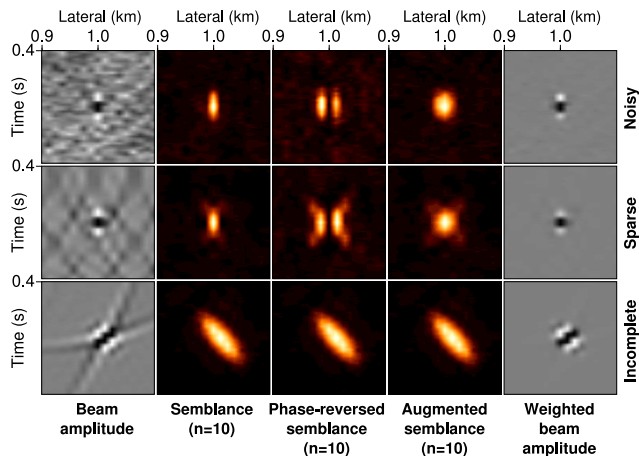

**Figure 4.** Close-up of reconstruction results for the left scatterer of the synthetic test introduced in Figure 3. The augmentation of a phase-consistent and a phase-reversed version of each considered coherence measure leads to a reconstruction that is slightly less resolved laterally, but proves to be insensitive to a sign change in amplitude for diffraction off asymmetric objects like edges. In addition, the normalized semblance norm can be used as a fully data-derived amplitude weight to suppress artefacts or noise in migrated images.

weak residual energy remaining for the conventional beam energy and the semblance norm. Both, the 10-th root version of the beam energy and the 10-th root semblance lead to diffraction focusing results of consistently high quality in all evaluated scenarios. Following from this systematic analysis and because its normalized character that favours weak and incompletely

sampled wavefields, we conclude that the $n$-th root semblance can be expected to be the most robust candidate for diffraction imaging with imperfect data. The strong suppression of reflected energy furthermore suggests that even without competitive diffraction separation first diffraction-enhanced images might be gained in moderately reflective media. Very similar to the anisotropic radiation characteristics of passive seismic sources, for diffraction off edges and structural steps, as they likely also occur at fault zones, the polarity of diffractions can change near the apex, which would lead to a bi-modal reconstruction. To

account for this radiation pattern, the conventional coherent focusing result can be augmented with its phase-reversed version (Figure 4). The augmented counterparts of the beam energy and the semblance norm are insensitive to the occurrence of a phase reversal, resulting in a more stable and only slightly less resolved reconstruction. Additionally, it can act as a data-driven migration weight to suppress artefacts in Kirchhoff-type diffraction focusing.

### 3.3 Imaging by projection

An alternative to diffraction focusing constitutes imaging by projection. The main mindset underlying this second strand of diffraction imaging is to start investigations in data space and to use the extracted information to arrive at an image. In coherent diffraction imaging, coherence analysis is carried out in the data domain to locally enhance and physically characterize waveform similarities that can be exploited for imaging. The previously discussed coherence measures can be readily employed. However, instead of using the reconstruction-centred image point parametrisation, the emerging diffracted wavefront is locally





characterized along the entire event, resulting in a transformation of the data volume into coherence and wavefront attribute
maps. A data-centred 2D analogue to the double-square-root equation (6) can be expressed in terms of the local tilt angle $\alpha_0$
and curvature radius $R_0$ of the wavefront as it is observed at location $x_0$ on the acquisition surface

$$\Delta t_{\text{diff}}(x_0, t_0) = \sum_{i=s,g} \frac{\sqrt{R_0^2 + 2\,R_0 \sin\alpha_0 \Delta x_i + \Delta x_i^2} - R_0}{v_0}, \tag{7}$$

with $\Delta x_i = x_i - x_0$ and $\Delta t_{\text{diff}} = t_{\text{diff}} - t_0$ (Höcht et al., 1999; Schwarz and Gajewski, 2017b). It is important to note that in
contrast to the process of image formation by focusing, here, the discussed coherence arguments are evaluated within a local
data aperture and assigned to the central data point within this aperture. So in contrast to equation (6), $(x_0, t_0)$, just like the
summation process itself, now also resides in data space. The actual reconstruction then consists of a subsequent mapping of
every locally coherent data point to a point in image space. In analogy to focusing-based time migration, analytical mapping
equations can be gained, by evaluating the stationary point $(x_{\text{apex}}, t_{\text{apex}})$ corresponding to the apex position of each individual
local diffraction traveltime curve. For expression (7), the projection corresponds to a mapping from each data point $(x_0, t_0)$ to
$(x_{\text{apex}}, t_{\text{apex}})$ via

$$x_{\text{apex}} = x_0 - R_0 \sin\alpha_0 \,, \tag{8a}$$

$$t_{\text{apex}} = t_0 + \frac{2\,R_0}{v_0}(\cos\alpha_0 - 1) \,, \tag{8b}$$

where $v_0$ denotes the locally constant near-surface velocity. Generally, by convention, we refer to near-surface quantities,
i.e. quantities that relate directly to the registration, with the subscript 0. Equations (8) in conjunction with expression (7) and
one of the positive-valued coherence measures can be used to set up an optimization problem to arrive at an approximate but
fully data-driven reconstruction of the subsurface scatterer distribution (Fomel, 2007; Schwarz et al., 2014; Bonomi et al.,
2014). In addition to considering a near-surface projection, wavefront slopes and curvatures can also be estimated using the
assumption of an effective replacement medium, which, like in equation (6) is defined by the root-mean-square velocity. A
so-called osculating equation connecting the near-surface projections and effective medium properties was first established by
de Bazelaire (1988) and generalized by Schwarz and Gajewski (2017b). For the sake of simplicity, only the 2D versions of the
data-centred diffraction moveout (7) and the projection equations (8) are given here. However, in a similar way, one can arrive
at corresponding 3D analogues of equations (8) by evaluating the stationary point of the 3D traveltime moveout expression.

In Figure 5 the maximized coherence for every data point as well as the corresponding projection results for the same
synthetic test as in Figure 3 and Figure 4 are presented. While all four considered coherence measures perform equally well
when the mere detection of coherent energy is concerned, noticeable differences can be observed with regards to overall
strength and the handling of interference. Both versions of the beam energy turn out to be not affected by the presence of
the strong conflicting reflected event, whereas semblance, due to its intrinsic normalization, reveals to be suited to estimate
the coherence of weak and energetic arrivals equally well. In contrast to the results gained with the conventional definition of
the semblance ($n = 1$), the 10-th root version does not suffer from interference effects and, similar to the preceding focusing
analysis, performs consistently well for all data points. Although all four displayed coherence measures can be used as a cost




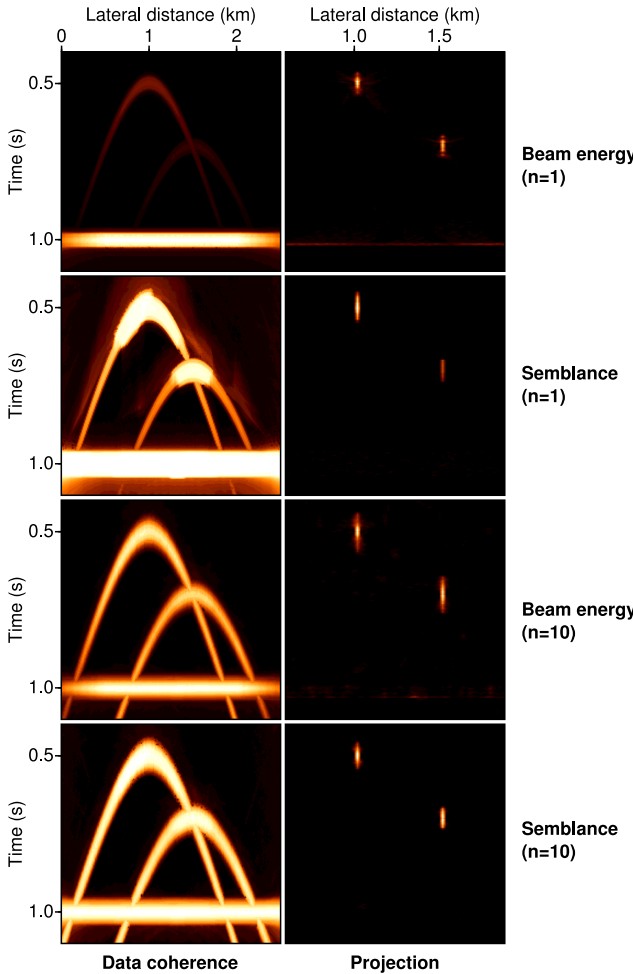

**Figure 5.** Results of data-driven coherence analysis (left) and subsequent projection (right) carried out for the same synthetic test case as in Figure 3. All measures detect coherence equally well, but reveal different characteristics when the overall strength and interference effects are considered. The reconstructions correspond to a count of mapped coherent amplitudes per image point.

function in data-driven coherence analysis, the use of normalized quantities allows for efficient and intuitive thresholding in subsequent processing steps, which, again, lets the 10-th root semblance appear as the most favourable candidate. Similar to coherent focusing, reflections are naturally suppressed by being projected in a diffuse manner.

As will be commented on in more detail in the discussion section, the data-centric nature of projection favours automation and macro-model-independent imaging. However, a major difference between data and image space constitutes the occurrence of intricate interference phenomena, in which multiple wavefields conflict with each other. This can be viewed as an efficient means of compression, but the decoding (i.e. the separation) of interfering wavefield components remains a challenging and computationally demanding task (compare the degradation of the conventional semblance norm in Figure 5). In contrast to





that, the image-centric character of coherent focusing naturally avoids these complications. For this reason, without loss of generality, the challenging data examples considered in the following section will all be based on coherent focusing.

## 4  Applications

The occurrence of diffraction phenomena is linked to the pre-dominant wavelength. Consequently, just like faults and fractures themselves, diffractions can be observed on essentially all scales probed in geophysical investigations of the upper crust. As with conventional Kirchhoff migration, there exist natural limitations of the suggested strategies for coherent diffraction imaging. However, with a range of ambitious field data examples we seek to demonstrate that rich diffracted wavefields exist in essentially all datasets and become assessable with the presented robust coherence arguments. Following the quantitative analysis of the different coherence measures discussed in the previous section, the images presented in the following were without exception generated with the 10-th root semblance norm. It may however be noted that the other coherence measures might have led to reconstructions of comparable quality. With exception of the 3D seismic land data example, all diffraction images were formed by augmenting sub-images with and without radiation pattern correction. Despite the depiction of the generalized workflow for coherent diffraction imaging in Figure 2, the presented images did not experience any image processing, but relied on a preceding separation step to suppress reflected energy from the considered pre- and poststack data (Schwarz, 2019b). While ground-truth reconstructions were not available, a systematic comparison with Kirchhoff-migrated images is provided for some of the examples.

### 4.1  Multi-channel seismic imaging off-shore Israel

The first data example consists of a marine multi-channel seismic field dataset acquired by TGS in the context of hydrocarbon exploration in the Levantine Basin, Eastern Mediterranean (e.g. Reiche et al., 2014). In contrast to the other data examples, this dataset includes source-receiver offsets of up to more than 7 km, which makes it well suited for conventional migration and inversion. The captured geology off the coast of Israel is primarily dominated by the presence of salt and connected processes in the upper crust (Krijgsman et al., 1999; Gradmann et al., 2005; Netzeband et al., 2006). Most notably, a large, laterally elongated salt body is overlain by vast sedimentary complexes accompanied by chaotic sequences in the shallow layers, possibly related to turbulence. Figure 6 compares a portion of prestack Kirchhoff migration with the corresponding coherent diffraction image for the geological units directly above the salt body. In both reconstructions three seismically distinct units can be identified. The uppermost unit consists of largely horizontal strata related to recent, mostly unperturbed sedimentation. Despite its overall smooth appearance, this sediment package contains faint signatures of channel-like structures that might have been caused by ocean currents eroding the sea floor. In the diffraction image this very reflective uppermost unit is almost entirely transparent, while the weak signatures of erosion are distinctly visible features.

   The opposite holds for the central unit, which appears as a largely transparent body with weak, chaotic internal structure. Again, like for the uppermost layers, the diffraction image indicates pronounced internal complexity, while mild reflective signatures vanish entirely from the reconstruction. Whereas in the migrated image the vertical extent and lateral complexity





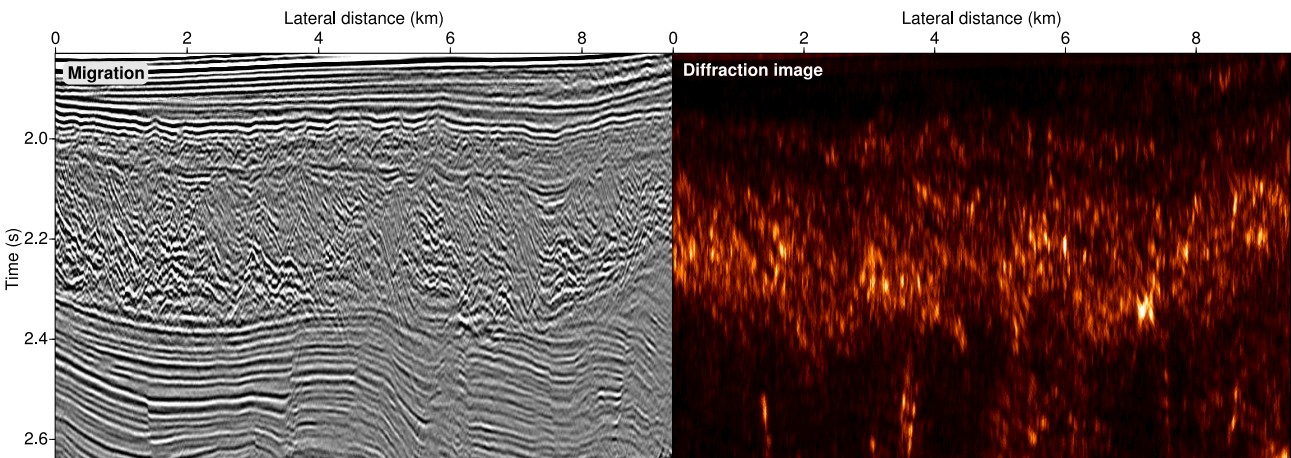

**Figure 6.** Comparison of close-ups of Kirchhoff migration (left) and the corresponding coherent diffraction image (right) for the industrial multi-channel data acquired in the Levantine Basin off-shore Israel in the Eastern Mediterranean. Both images are of complementary nature in that the migration highlights predominantly horizontal features related to sedimentation, whereas the diffraction-based reconstruction emphasizes small-scale structural complexity related to dynamic processes like turbulent erosion or faulting connected to salt tectonics.

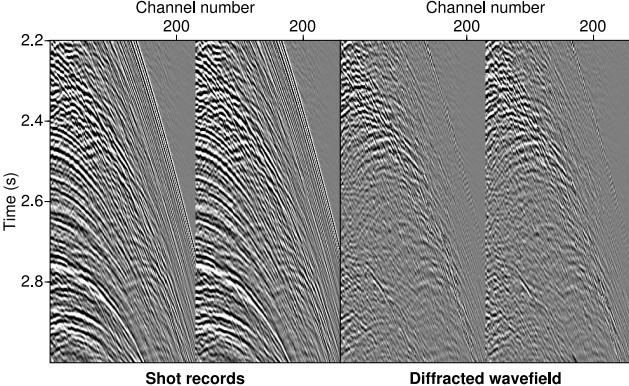

**Figure 7.** Two exemplary common-source gathers highlighting the effectiveness of the preceding wavefield separation step as well as the presence of a rich diffracted wavefield which became assessable in the prestack domain. Amplitude strong diffraction at about 2.4 s can be connected to the lowermost part of the central chaotic unit which reveals pronounced internal small-scale complexity.

remain largely obscured, the diffraction image favours a clear delineation of this complex unit and its internal structure. Figure 7 shows close-ups for two exemplary common-source gathers, in which a multitude of diffractions related to the lower part of this chaotic unit can be observed in the separation. The lowermost unit reveals primarily horizontally stratified sedimentation which is disrupted by several faults that were caused by salt-related tectonics. Again, the diffraction image is of complementary value in that it highlights geological features related to dynamic processes, while the reflection-dominated conventional migration





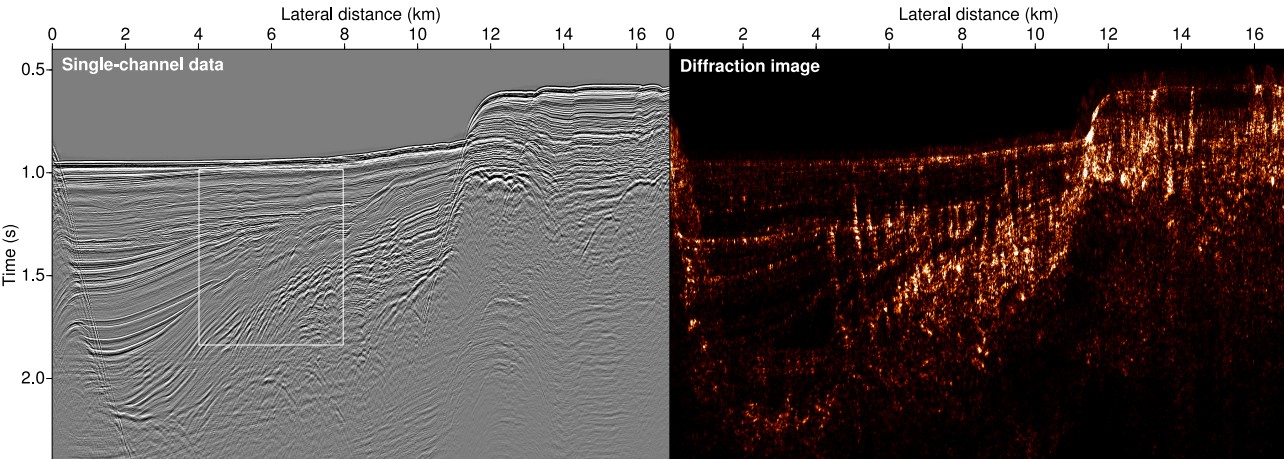

**Figure 8.** Excerpt of the unprocessed input data of a marine seismic single-channel acquisition carried out near Santorini in the Aegean Sea (left) and the result of coherent diffraction imaging (right). Intricate fault and fracture networks are revealed, as their presence causes complicated wave diffraction. Compare Figure 9, where the close-up indicated by the white frame is investigated in more detail. Aside from faulting, erosional unconformities and the interface between sediments and the crystalline basement are well-recovered.

emphasizes sedimentary features indicative of more stationary episodes in Earth's history. For more details on the interpretation of the data and the captured geology, we refer to Gradmann et al. (2005) and Reiche et al. (2014).

### 4.2 Single-channel seismic imaging in the Aegean Sea

As a second example we consider a single-channel marine seismic dataset that was acquired near the island of Santorini in the Aegean Sea (Hübscher et al., 2015; Nomikou et al., 2016b). The wider geological setting includes the Anydros Basin – a region known to be shaped by extensive volcanism resulting in pronounced structural complexity. Owing to past and ongoing tectonic processes, the upper crust is disrupted by major fault and fracture networks. It is dominated by the Kolumbo submarine volcano, whose activity might have triggered devastating Tsunamis in the past (Nomikou et al., 2016a). Despite the fact that

only a single channel was available, the data can be considered of reasonably high-quality. Owing to a short shot interval, the near-offset data set provides dense spatial sampling, which is deemed ideal for high-resolution diffraction imaging. Figure 8 shows the single-channel data before processing together with the reconstruction based on coherent diffraction imaging. Captured is a larger sedimentary basin near the flank of the Kolumbo submarine volcano.

In its unprocessed form the dataset is dominated by reflected energy and only occasionally, small-scale structural complexity

is indicated by faint interference patterns. In strong analogy to the marine seismic multi-channel dataset discussed before, the diffraction image appears contrast-rich and highly resolved predominantly in regions where dynamic processes were at work, whereas sedimentary reflections are fully transparent in the reconstruction. Similar to the previous example the dominant features are connected to a major fault system in the right part of the sedimentary basin. This intricate network is thought to represent a major flower structure, which is connected to the rifting regime it is embedded in (Hübscher et al., 2015). Likewise





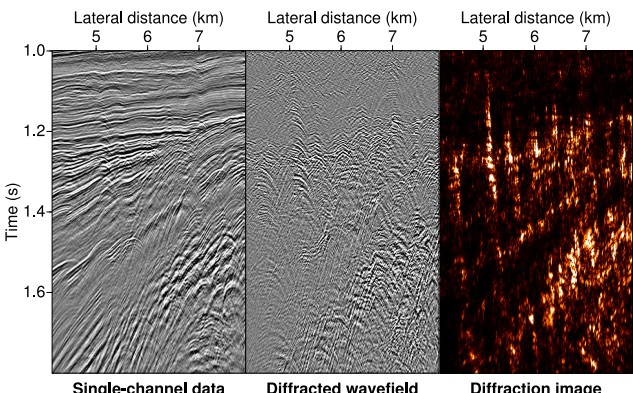

**Figure 9.** Comparison of the unprocessed single-channel data (left), the result of diffraction separation (centre), and the diffraction-based reconstruction (right) for the close-up indicated by the white frame in Figure 8. Coherent focusing of the separated wavefield enables a clear delineation of individual faults with a lateral separation as small as 200 m.

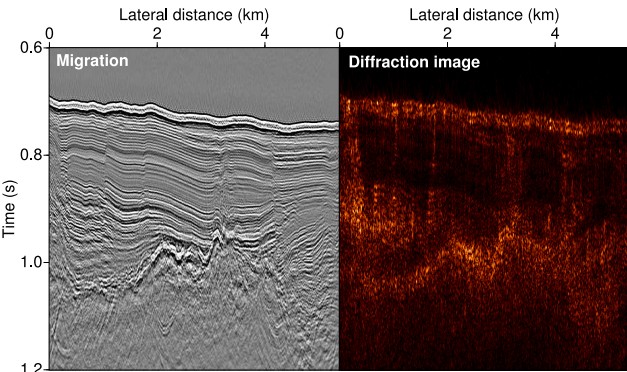

**Figure 10.** Comparison of the Kirchhoff-migrated image with the result of coherent diffraction imaging for a small excerpt of an adjacent portion of the line. Like in Figure 9, a lateral structural resolution of up to approximately 200 m was achieved, which lets a set of quasi-parallel faults become distinguishable in the lower rightmost part of the diffraction image.

well-resolved are erosional unconformities and the sediment-crystalline-basement interface which are of rugged character with small-scale lateral complexity. To closer inspect the success of the coherent diffraction imaging workflow, Figure 9 compares a close-up (indicated by the frame in Figure 8) of the reflection-dominated input data, the result of diffraction separation and the diffraction-based reconstruction. Quasi-parallel faults are individually recovered with a lateral separation as small as 200 m, which is broadly in the range of the predominant seismic wavelength. To further illustrate the lateral resolution achievable with

diffraction imaging, Figure 10 compares an excerpt of a Kirchhoff-migrated image with the corresponding diffraction-based reconstruction. In the latter, aside from the contrast-rich detection of an internal unconformity and the top of the crystalline basement, small-scale sub-parallel faults are recovered at lateral distances above 4 km. Again, the achieved lateral resolution approaches the order of the seismic wavelength, thereby highlighting the high-resolution potential of diffraction imaging. For


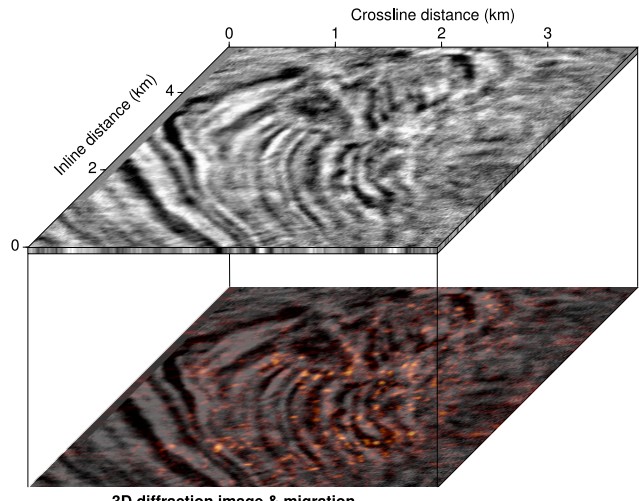

**Figure 11.** A slice of the 3D migration cube that captures a portion of the Stratton gas field in Southern Texas (top) and the corresponding augmented image combining the migration with a semi-transparent version of the the coherent diffraction image (bottom). Highly resolved features of several fluvial channel systems known to exist at reservoir level are described well by the diffraction image (compare Hardage et al., 1994).

a more detailed geological interpretation and an example of diffraction-based velocity model building in depth, we refer the
reader to Preine et al. (2020).

### 4.3 3D seismic imaging in Southern Texas

Complementing the two marine seismic examples, here we will briefly demonstrate the successful application of 3D diffraction imaging on land. In 2014 Bob Hardage and Scott Tinker of the University of Texas at Austin decided to make the *Stratton* 3D dataset, consisting of a migrated reference volume, unprocessed prestack gathers, vertical seismic profiling data, well log infor-
mation, and other related resources freely available to the research community. The multi-channel seismic data were recorded in four overlapping swaths, which each consisted of six receiver lines separated by approximately 400 m. The acquisition was intended to capture a portion of the Stratton gas field located in Southern Texas, with the aim of better understanding the internal architecture of complicated oil and gas reservoirs. As the gas field is of fluvial origin, dominant structural features include overlapping fluvial channels (Hardage et al., 1994). To illustrate the potential added value of coherent diffraction
imaging for 3D seismic interpretation, Figure 11 shows an exemplary migration slice at reservoir level and compares it with an augmented image consisting of the same migration slice overlain by a semitransparent version of the coherent diffraction image. As expected from a thin-bed fluvial reservoir system, the diffraction-based reconstruction indicates a high degree of structural completely. A comparison of the migration with the augmented diffraction image reveals several spatially coherent diffractive corridors that follow established channel trends and might also be indicative of minor faulting, which is of potential
significance for the monitoring of fluid flow.





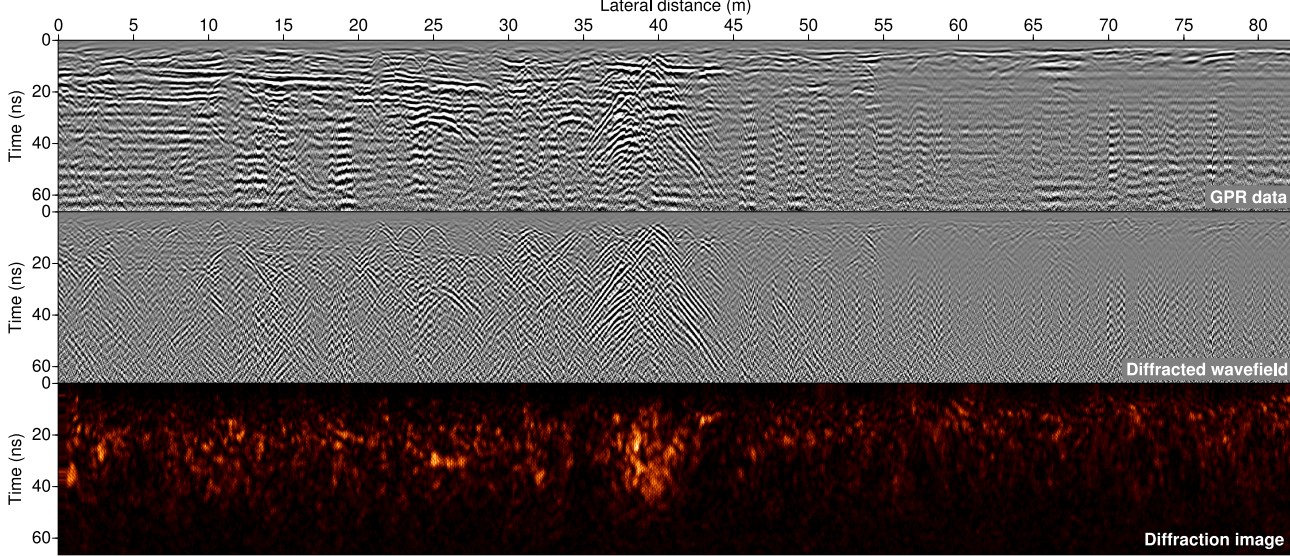

**Figure 12.** Ground-penetrating radar (GPR) line capturing the uppermost subsurface near the shore of Long Beach Island, New Jersey, which was acquired in a multi-disciplinary effort to study the impact of seasonal tropical storms on coastal dynamics and the connected structural changes of the environment. Displayed are the input data (top), the result of diffraction separation (centre) and the coherent diffraction image (bottom), in which a strongly diffractive V-shaped structure can be delineated.

## 4.4  GPR imaging on Long Beach Island

The fourth and final example consists of a ground-penetrating radar (GPR) line that was acquired by the US Geological Survey in the frame of a multi-disciplinary effort to study the impact of seasonal tropical storms on coastal change as part of the Barrier Island and Estuarine Wetland Physical Change Assessment (Zaremba et al., 2016). The campaign was a response to Hurricane

Sandy which approached the US East coast in October 2012. A connected storm surge and wave activity caused major alterations of the shore line, resulting in a modified coastal topography, geology, and hydrology with immediate impact on regional ecosystems. The GPR measurement campaign was carried out roughly three years after Hurricane Sandy hit the shore and, in conjunction with remote sensing and sedimentological observations, had the main goal of systematically assessing physical changes of the coast induced by seasonal storms in order to update systemic models to improve predictability (Plant et al.,

2018). In Figure 12 the exemplary line of the GPR data volume is displayed with the result of diffraction separation and the outcome of coherent diffraction focusing. Along the line, pervasive diffraction can be observed which indicates a strong degree of structural complexity near the surface. At the centre of the line a V-shaped structure is revealed to be responsible for pronounced electromagnetic scattering, suggesting the presence of a strong material contrast. Similar to the single-channel seismic example, owing to the fact that emitting and receiving antennae typically coincide, reflections in GPR data are predominantly

sensitive to vertical structural changes, whereas lateral information stemming from channels and other dynamically important erosional structures is largely encoded in the diffracted wavefield.





## 5   Discussion & outlook

In line with previous works on diffraction imaging, the systematic synthetic investigations and, in particular, the range of considered challenging field data examples suggest that diffraction imaging bears the potential to shed a unique light on

intricate fault and fracture networks and other dynamically relevant features. As indicated in the generalized workflow shown in Figure 2, coherent diffraction imaging lets us arrive at resolved images of lateral discontinuities, which, in some sense, encode the dynamic history and past and present stress states of the crust. In conjunction with conventional reflection-dominated images resulting from migration, diffraction images were shown to be of immediate and complementary use in interpretation (Burnett et al., 2015; Preine et al., 2020). To decipher the signatures of faults and fracture systems in diffraction images is

of immediate importance for the construction of geodynamic models or the simulation and assessment of fluid flow (Sibson, 1994). The suggested incorporation of coherence arguments in constructing images is expected to aid in this interpretation task and interface well with automation strategies that build on machine learning techniques that have their origin in computer vision (e.g. Wu et al., 2019). Generally it can be argued that the positive-valued character of the coherent reconstructions favours the subsequent application of useful tools from image processing.

Aside from arriving at high-resolution structural subsurface reconstructions, diffractions also provide unique illumination in various important data sub-domains, like the zero-offset (single-channel), common-source, or common-receiver configuration. Illustrated by the marine single-channel example, the 3D reconstruction based on reduced land data, and the zero-offset GPR application, diffractors can indeed be viewed as structure-related passive sources, which suggests a systematic use of cost-effective, reduced acquisitions in seismic investigations (Schwarz, 2019a). In addition to the potential of lowered acquisition

costs, the strong similarity of diffracted and passive events also suggests continuing technological transfer between controlled-source and passive-source seismology (Li et al., 2020). The introduction of interference-sensitive data abstractions like the considered sign-sensitive n-th root was shown to have a stabilizing effect on diffraction imaging and is also expected to benefit automated coherence analysis as a whole.

Although not considered here, it should be stated that, similar to the employment of ray tracing or eikonal solvers to arrive

at more accurate focusing trajectories in laterally complex media, the projection recipe can likewise be extended to account for more complicated and demanding scenarios. In fact, wavefront tomography builds on the same attribute fields and coherence maps as were fed to the analytic mapping equations (8), but makes use of dynamic ray tracing to perform the subsequent projection into image space. Also, the method was shown to reliably utilize diffracted energy to arrive at a resolved estimate of scatterer locations in depth (Duveneck, 2004; Bauer et al., 2017). While the projection step, like in focusing, could be

performed for a pre-defined velocity distribution, projection was shown to lend itself well for a largely automated reconstruction of not only the scatterer locations, but also the macro-velocity structure in an iterative process. As was demonstrated e.g. by Bauer et al. (2019), data-centric mapping lets one tag and track the contribution of each data point into the image, which provides a powerful interface to machine learning techniques, commonly used for image segmentation tasks (Shustak and Landa, 2018). In conjunction with time-reversal imaging (e.g. Mosk et al., 2012), which can likewise be considered a projection

technique, coherence arguments and wavefront tomography were recently demonstrated to form a powerful framework for the





joint inversion of passive-source and medium properties (Diekmann et al., 2019). Sufficiently dense spatial sampling provided, coherent diffraction imaging, in particular when phrased as a projection problem, is expected to be applicable to passive seismic data, as the problem strongly resembles diffraction imaging in the common-source domain (Schwarz, 2019a). Differential semblance optimization and, more broadly, migration velocity analysis can be viewed as the focusing-based counterpart of

wavefront-tomographic inversion and might likewise be investigated in the future.

It needs to be emphasized that, in particular when only reduced datasets like single-channel volumes are acquired, diffractions bear a distinct advantage over reflections, in that their illumination is encoded in various sub-domains of the multi-channel response (Schwarz, 2019a; Preine et al., 2020). However, it must also be noted that the process of diffraction is inherently three-dimensional, which can cause out-of-plane energy to contaminate the data with the potential to result in the occurrence

of artefacts in the reconstruction. Although coherence arguments were demonstrated to help with handling data imperfections, the highest-quality reconstructions are expected for sufficiently dense spatial data sampling. With the rise of large-N arrays and, in particular, the emerging new data resource of distributed fibre-optic strain sensing, wavefields start to be acquired quasi-continuously, which is expected to extend the reach of coherence analysis and diffraction imaging (Jousset et al., 2018).

## 6   Conclusions

We have presented a simple yet powerful framework to arrive at highly resolved structural images of the upper crust by making use of the diffracted component of the wavefield. By means of controlled synthetic test cases, we introduced and systematically investigated four positive-valued coherence measures which find strong correspondence in visual perception. Based on the prerequisite of reasonably dense spatial sampling, we suggested a generalized workflow for diffraction imaging, in which image formation is either achieved by focusing or by projection of coherent contributions. While synthetic tests suggest the overall

robustness of coherence-based diffraction imaging, the investigation of seismic and ground-penetrating radar data acquired on land and in the marine environment emphasize the applicability and complementary nature of diffraction imaging for a broad range of geophysical applications. Owing to its high-resolution potential, the presented workflow helped to delineate small-scale structural features such as fluvial channels, erosional unconformities, and intricate fault and fracture systems, which remain challenging to image by conventional means.

*Code and data availability.*   The two suggested strategies for coherent diffraction imaging were implemented in the emergent high-level language Julia. Both, the codes and intermediate and final results are available upon reasonable request.

*Author contributions.*   BS performed the computations and is responsible for the theoretical aspects of the paper. CMK helped strengthening the overall scope and added to interpretational aspects and the discussion of the presented results. Both authors contributed equally to proofreading and the final preparation of the manuscript.





*Competing interests.* The authors declare that they have no competing interests.

*Acknowledgements.* Important strands of the presented research were supported by Geo.X, the Research Network for Geosciences in Berlin in Potsdam. The academic single-channel and the industrial multi-channel datasets were acquired and kindly shared by Christian Hübscher and TGS. We are grateful to the US Geological Survey and the Bureau of Economic Geology at the University of Texas in Austin, who collected the Stratton 3D seismic field data and the investigated GPR line from New Jersey and made these valuable resources available to

the public. Wavefield simulations for the synthetic examples as well as pre-processing and visualisations were carried out with Seismic Unix, whose contributors we wish to acknowledge. We also warmly thank Alexander Bauer and Jonas Preine for continued stimulating discussions.



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
