# Peer review of "Coherent diffraction imaging for enhanced fault and fracture network characterization"

_Solid Earth, 2020_

## Referee Comment (RC1) · Anonymous Referee #1 · 22 Jun 2020

Coherent diffraction imaging for enhanced fault and fracture network characterization Benjamin Schwarz1 and Charlotte M. Krawczyk1,2 1GFZ German Research Centre for Geosciences, Albert-Einstein-Str. 42-46, 14473 Potsdam, Germany 2Technical University Berlin, Ernst-Reuter-Platz 1, 10589 Berlin, Germany Correspondence: Benjamin Schwarz (bschwarz@gfz-potsdam.de)

Review General: The authors present data examples of the application of diffraction imaging based on focussing with unweighted time migration including coherence measurement methods along the migration travel time surfaces without going deep into the mathematical theory. As introduction synthetic data sets illustrate the diffraction

phenomena and its relation to reflected signals. They explain the difference between focussing followed by coherency application in the image domain and coherency application in the data domain followed by projection. Coherency methods are discussed in detail in the image domain and also an application in the data domain followed by projection is compared to the same 2D synthetic dataset. A short introduction into the projection method with multiple wave front attributes is given with most relevant literature references for further reading. The real data examples are all applied by focussing followed by coherency application which is the simples form of diffraction imaging. An 2D multichannel dataset was used additionally to show the wavefield separation of reflections and diffractions in the shot-gather domain as a pre-processing stage and compared the pre-stack time migration and the migrated diffracted image. From a 2D single channel profile the diffracted image was compared to the zero-offset section and the separated diffracted zero offset section. From the same dataset additionally, a post stack time migration and the diffracted image was used to characterize the subsurface structure. A time slice of a 3D migrated land dataset data and the corresponding diffracted image time slice very nicely show additional information how diffracted images can contribute to an interpretation. The last example is an application to ground penetration radar with a zero-offset section, the separated diffracted wavefield and the diffracted image. For all data examples references are given for further reading.

Specific Comments: The comment here I make in the following could be the topics in the discussion helping people without deep theoretical understanding (application paper) when to use the focussing method and how to interpret the diffracted image although you already mentioned the topics in a broad discussion. The paper nicely shows data examples of diffracted images and how use them in conjunction with the reflective image to characterise the subsurface. But what I am missing are the uncertainties and limitations of the method. Because diffractions are generally 3D a 2D profile will also show side diffractions. The hope is that the presented coherence method will cancel most of the events, but this has not been shown, most critical I see here the single channel data. Maybe additionally a simple guideline can help nontheoretical readers to understand what of information could be expected for this kind of focussing application depending on the input information: velocity knowledge, single channel/multichannel, type of diffractions (generated by fault zones - edge wave with polarity change, point/volume diffractions without polarity change) apparent velocities, and time/depth errors from side echoes. Velocity knowledge: if multichannel data exist, the application of the velocity estimated from MCS reflections seems to be the most intuitive and hopefully the coherency will cancel the diffractor images which are generated from side echoes. A velocity estimation only from diffractions along a 2D profile would I not expect to be to accurate. I think all of the problems I mentions can be solved for 3D data with a multi-attribute analyses followed by projection which seems to get very powerful method in the future.

Individual Correction / Comment: Page 3 Line 71 Correction/Comment: . . . when a wavefield encounters a relevant property change (e.g. that has a local curvature) of or below the wave length . . . Comment: a horizontal interface with an impedance anomaly will also create diffractions.

Page 6 Figure 2 Correction/Comment: Left side: Focussing Section 3.2, Coherence measurement Section 3.1. Right side: Coherence measurement Section 3.1, Projection Section 3.3 Comment: not Section 2.x

Page 10 Figure 4 Comment: not clear what Phase-reversed semblance and augmented semblance means (I have some idea), but please reference to your equation what you did when.

Page 10 Line 219 Correction/Comment: the polarity of diffractions can change near the apex for zero offset data, which . . .. Comment: for 2D offset data the phase change occurs along the boundary ray (e.g. edge diffractions). This position can be found e.g. by a double diffraction stack (vector stack) during the migration in a shot-gather (the stationary point / tangential alignment). Side echoes may have no phase change at all.

Page 13 Line 299 Correction/Comment: which appears as a largely (transparent) body

with (weak), chaotic internal structure. Comment: it does not look transparent or weak with this gain applied to the section.

---

## Referee Comment (RC2) · Anonymous Referee #2 · 9 Jul 2020

General comments: The authors has presented two diffraction imaging approaches based on the assessment of coherence to detect geological features with small-scale complexity. One of the techniques is coherent focusing in which wavefield coherence is assessed in image space. Whereas in the second one, which is called coherent projection, first data coherence is evaluated, then the back-projection is conducted. Inspired from the field of optics, the proposed methods use intensities to evaluate coherence and reconstruct the image. The authors have declared that the coherent projection approach is not practical due to difficulty in separating wavefield components as well as being computationally heavy. Hence, they tested the focusing method on few real data including a multichannel and a single channel seismic data, a 3D seismic data,

and a GPR data for imaging the upper crust. The results confirmed the success of the technique to detect faults and fracture networks, which later on can be implemented in qualitative interpretation. Generally, the article structure is clear and the content is rich. The concepts and techniques, presented in this paper, are novel, and they are supported with enough successful real data examples.

Specific comments: Please find my comments and suggestions below:

1. In page 3, line 86, the authors claim that since diffractions do not follow Snell's law, and they always have similar shape, therefore, diffracted signals are often an order of magnitude weaker than their reflected counterparts. Please elaborate on this matter and explain why the diffracted signals are often only an order lower than the reflected signals?

2. Page 6, figure 2: please check the section numbering. I recommend to add a section referencing in the figure caption, for instance: While projection-type imaging schemes start directly in data space (refer to section 3.3), focusing techniques typically are image-centred (refer to section 3.2).

3. What is the criteria to define the n-th order of the beam energy and the semblance? Does the algorithm utilize any optimization procedure to find the optimum order? Please elaborate on this matter in the last paragraph of page 8.

4. Page 10, figure 4: please explain whether augmentation is applied on the phase reversed semblance, or on the semblance directly? Besides, the explanation in the text about the algorithm and order of applying both phase reversing and augmentation is not clear.

5. Do the diffraction images, obtained via the coherent focusing technique, have the capability to be employed for future quantitative interpretation purposes?

6. Page 17, line 347: what do the authors mean with "a high degree of structural completely"? Please check the sentence.

---

## Editor Comment (EC1) · Roger Soliva (Editor) · 17 Jul 2020

Dear authors,

We now have the feedback from two reviewers suggesting moderate revision of the manuscript. I encourage you to revise the mnuscript as suggested by the reviewers and answering to their comments point per point.

Best regards, Roger Soliva

---

## Author Comment (AC1) · 2 Aug 2020

Dear referee,

many thanks for a very positive and encouraging review. We take your comments very seriously and will make sure to fully honour them during the revision of the manuscript. In the following, we provide you with our personal view on the issues you have raised and also indicate, how we intend to alter the manuscript to reflect the desired changes. For convenience, your individual comments appear bold, replies remain in standard font and intended textual modifications/additions are indicated by italic letters.

**The results confirmed the success of the technique to detect faults and fracture networks, which later on can be implemented in qualitative interpretation. Generally, the article structure is clear and the content is rich. The concepts and techniques, presented in this paper, are novel, and they are supported with enough successful real data examples.**

We are happy that you see merit in the proposed methodological framework and that you found the scale-spanning data examples convincing. Thanks for the encouraging feedback.

**1. In page 3, line 86, the authors claim that since diffractions do not follow Snell's law, and they always have similar shape, therefore, diffracted signals are often an order of magnitude weaker than their reflected counterparts. Please elaborate on this matter and explain why the diffracted signals are often only an order lower than the reflected signals?**

We realise that the way we have formulated these sentences might cause confusion. Instead of "*It is interesting to note that, in contrast to reflections and other wavefield components obeying Snell's law, diffractions always have a similar shape, no matter which data configuration is considered. This implies that diffractions not only provide improved illumination and encode highly resolved information on the structures that caused them, but it also explains why diffracted signals are often an order of magnitude weaker than their reflected counterparts.*" we will include the following sentence: "*In contrast to reflections, diffractions are not constrained by Snell's law and, thus, radiate uniformly in all directions. As a result, diffracted wavefields provide improved illumination and encode highly resolved structural information, but also rapidly decay with increasing distance from the scatterer.*"

**2. Page 6, figure 2: please check the section numbering. I recommend to add a section referencing in the figure caption, for instance: While projection-type imaging schemes start directly in data space (refer to section 3.3), focusing techniques typically are image-centred (refer to section 3.2).**

Thanks for noticing this mistake and for the good suggestion, we will correct the section numbering and will explicitly refer to the imaging sections via: "*While projection-type imaging schemes (Section 3.3) start directly in data space, focusing techniques (described in Section 3.2) typically are image-centred.*"

**3. What is the criteria to define the n-th order of the beam energy and the semblance? Does the algorithm utilize any optimization procedure to find the optimum order? Please elaborate on this matter in the last paragraph of page 8.**
This is quite an interesting remark and we fully agree that this could be optimized. However, we also found by means of trial and error in a variety of different scenarios that for values larger than n=10, changes become very subtle and barely distinguishable. Like in earthquake seismology, taking the n-th root leads to an equalization of different amplitudes with the benefit that weak and strong contributions are treated equally in the analysis. Following your advice we now include the following brief discussion in the last paragraph of Section 3.2: "*Taking the n-th root of the amplitude as suggested in expression (5) has the effect of making coherent arrivals of different strength more comparable. While the suggested value of 10 results from experience with a variety of data configurations, it can be shown that this equalization in amplitude typically saturates for a reasonably low n already. In principle, the problem of finding a suitable root order could be phrased as an optimization problem, driven by the amplitude content of the data. However, a fixed value of 10 was shown to be successful in bridging several orders of magnitude and, therefore, is deemed a reasonable choice in most practical scenarios.*"

**4. Page 10, figure 4: please explain whether augmentation is applied on the phase reversed semblance, or on the semblance directly? Besides, the explanation in the text about the algorithm and order of applying both phase reversing and augmentation is not clear.**
You are right, we should have spent some more time on sufficiently explaining the "*augmentation*". In line with referee 1 who had very similar remarks we will include

the following additional sentences to describe the process (starting in line 220): "*Every data point is once treated as a potential stationary point at which an artificial phase reversal is performed before evaluating the coherence measure. Both results, the one gained without reversing the phase, and the one for which the phase is reversed, are compared and the higher value contributes to the augmented image.*"

**5. Do the diffraction images, obtained via the coherent focusing technique, have the capability to be employed for future quantitative interpretation purposes?**
You are right, we believe this is quite important to stress. To account for your question, we will conclude the discussion section with a dedicated paragraph on the implications of coherent diffraction imaging for (structural and quantitative) interpretation. Specifically, we will include the following sentences: "*The non-normalized beam energy (n=1) directly relates to the diffraction's focusing intensity, which is proportional to the square of the beam's amplitude and, therefore, to the strength of the impedance contrast at which diffraction occurred. On the contrary, higher-order versions of the beam energy (n>1) no longer deal with accurate, but rather distorted amplitude and phase information and, accordingly, cannot be used for quantitative interpretation. The same holds for the semblance norm in general, as its intrinsic normalization "evens out" amplitude discrepancies due to material contrasts of different strength. While all of the coherence measures suffer from the loss of phase information in the final reconstruction, the semblance coefficient, due to its normalization, can be used as a reliable weight for artefact and noise suppression in conventional wavefield focusing. The resulting images have higher quality yet largely preserve amplitude and phase information critical for quantitative geological interpretation of the imaged geology.*" In order to properly account for the importance of interpretational implications, we will rearrange and subdivide the discussion section into the following distinct subsections: "*5.1 Potential and extension of the method*", "*5.2 Limitations and challenges*" and "*5.3 Geological interpretation*" and additionally address attribute analysis in the following new paragraph: "*While the presented workflow discusses the best use of the physical information content of the recorded data through diffraction-targeted processing, structural interpretation makes*

*additional use of the growing amount of seismic attributes (Chopra and Marfurt, 2005; Barnes, 2016) – an integral approach of seismic interpretation aiming at mapping geological features. Like coherence (gained via cross-correlation of neighbouring traces in the reflection dominated migrated image), these attributes are often used on their own to improve the interpretation of fault structures. Alternatively, attributes can be assessed in combination or help in establishing cross-plotting maps (e.g. Endres et al., 2008; Lohr et al., 2008; Torrado et al., 2014; Wang et al., 2015). In this frame, coherent diffraction images can be viewed as physics-guided feature maps that naturally complement more conventional attributes commonly used for interpretation. To additionally foster the bridging from faults to fractures, data acquisition can likewise play an important role (see concept and example in Krawczyk et al., 2015). In near-surface applications in the field, using shear waves instead of compressional waves for seismic surveying has proven a powerful strategy for increasing structural resolution (e.g. Krawczyk et al., 2012; Beilecke et al., 2016). A combination of the proposed high-resolution imaging workflow with these new forms of data acquisition is expected to shed additional light on subsurface pathways, fault extent and fault connections in the subsurface, which are increasingly important for the assessment of structural integrity and fault behaviour or, ultimately, deformation monitoring in an area."*

**6. Page 17, line 347: what do the authors mean with "a high degree of structural completely" Please check the sentence.**

Thanks for pointing this out, the word "*complexity*" was accidentally misspelled.

On behalf of the authors,
Benjamin Schwarz

---

## Author Comment (AC2) · 2 Aug 2020

Dear referee,

we appreciate the time you have invested. We believe that your comments are well justified and thank you for the valuable feedback which will helps to improve the quality of the manuscript. In the following, we provide you with our personal view on the issues you have raised and also indicate, how we intend to alter the manuscript to reflect the desired changes. For convenience, your individual comments appear bold, replies remain in standard font and intended textual modifications/additions are indicated by italic letters.

[Figure]

**The comment here I make in the following could be the topics in the discussion helping people without deep theoretical understanding (application paper) when to use the focussing method and how to interpret the diffracted image although you already mentioned the topics in a broad discussion. The paper nicely shows data examples of diffracted images and how use them in conjunction with the reflective image to characterise the subsurface. But what I am missing are the uncertainties and limitations of the method. Because diffractions are generally 3D a 2D profile will also show side diffractions. The hope is that the presented coherence method will cancel most of the events, but this has not been shown, most critical I see here the single channel data. Maybe additionally a simple guideline can help non-theoretical readers to understand what of information could be expected for this kind of focussing application depending on the input information: velocity knowledge, single-channel/multichannel, type of diffractions (generated by fault zones - edge wave with polarity change, point/volume diffractions without polarity change) apparent velocities, and time/depth errors from side echoes. Velocity knowledge: if multichannel data exist, the application of the velocity estimated from MCS reflections seems to be the most intuitive and hopefully the coherency will cancel the diffractor images which are generated from side echoes. A velocity estimation only from diffractions along a 2D profile would I not expect to be to accurate. I think all of the problems I mentions can be solved for 3D data with a multi-attribute analyses followed by projection which seems to get very powerful method in the future.**

We thank you for this comment and we are convinced that some more detail on what to expect and more broadly on what could go wrong, will prove valuable information for the reader. We are aware of the principal complications that can arise from out-of-plane scattering and briefly address this in line 408: "*However, it must also be noted that the process of diffraction is inherently three-dimensional, which can cause out-of-plane energy to contaminate the data with the potential to result in the occurrence of artefacts in the reconstruction.*" It seems indeed worthwhile to more extensively elaborate on this

and other limitations. Following your advice, we will include the following additional sentences: "*Provided accurate velocity information is available, out-of-plane contributions in two-dimensional (2D) surveys are naturally suppressed, if the scattering structure is located reasonably far off the acquisition plane. However, less distant out-of-plane scattering can hardly be distinguished from valuable in-plane contributions, which is why 2D diffraction-based reconstructions must generally be assessed with care. This is particularly true for single-channel data, for which a reliable velocity model might not be available. In order to gain trust in diffraction-derived velocity information and coherent diffraction images the mere quality of focusing might be complemented by a joint assessment of the reflected wavefield. Powerful and reliable reflection-based velocity inversion schemes exist and can be used, if sufficient offset information is available. Thus, because reflected energy is less likely to stem from out-of-plane structures, the integrated interpretation of reflection and diffraction images can help to improve velocity models and identify off-plane scattering in 2D surveys (Preine et al., 2020). All these complications become superfluous for sufficiently dense three-dimensional acquisition strategies, which, therefore, are deemed ideally suited for reliable subsurface imaging with the diffracted wavefield.*" To better reflect the important issue you have raised and in order to increase readability we will rearrange and subdivide the discussion section into three distinct subsections ("*5.1 Potential and extension of the method*", "*5.2 Limitations and challenges*" and "*5.3 Geological interpretation*").

**Individual Correction / Comment: Page 3 Line 71 Correction/Comment: . . . when a wavefield encounters a relevant property change (e.g. that has a local curvature) of or below the wave length . . . Comment: a horizontal interface with an impedance anomaly will also create diffractions.**
You are right, but we would argue that this impedance anomaly itself represents a change in material properties and needs to be localized for wave diffraction to occur. Please let us know if we have a wrong perception of the scenario you are referring to, but like related works we specifically focus on non-Snell scattering, for which an anomaly is usually defined by means of a sufficiently high local curvature.

**Page 6 Figure 2 Correction/Comment: Left side: Focussing Section 3.2, Coherence measurement Section 3.1. Right side: Coherence measurement Section 3.1, Projection Section 3.3 Comment: not Section 2.x**

Many thanks for pointing this out – Figure 2 now shows the correct section numbers.

**Page 10 Figure 4 Comment: not clear what Phase-reversed semblance and augmented semblance means (I have some idea), but please reference to your equation what you did when. Page 10 Line 219 Correction/Comment: the polarity of diffractions can change near the apex for zero offset data, which . . .. Comment: for 2D offset data the phase change occurs along the boundary ray (e.g. edge diffractions). This position can be found e.g. by a double diffraction stack (vector stack) during the migration in a shot-gather (the stationary point / tangential alignment). Side echoes may have no phase change at all.**

You are right, the augmented version of the semblance deserves some more explanation. In fact, the double diffraction stack is exactly what we perform before augmentation. We will include the following sentences (in line 220): "*Every data point is once treated as a potential stationary point at which an artificial phase reversal is performed before evaluating the coherence measure. Both results, the one gained without reversing the phase, and the one for which the phase is reversed, are compared and the higher value contributes to the augmented image.*" In addition, we will replace the phrase "*near the apex*" with "*at the stationary point*" and specifically refer to the definition of the beam energy and the semblance norm (equations (2) and (4)).

**Page 13 Line 299 Correction/Comment: which appears as a largely (transparent) body with (weak), chaotic internal structure. Comment: it does not look transparent or weak with this gain applied to the section.**

We agree and will not refer to the respective layer as "*transparent*" or "*weak*".

On behalf of the authors, Benjamin Schwarz